# A LARGE-SCALE ANALYSIS ON METHODOLOGICAL CHOICES IN DEEP REINFORCEMENT LEARNING

## ABSTRACT

Deep reinforcement learning research has been the center of remarkable scientific progress for the past decade. From winning one of the most challenging games to algorithmic advancements that allowed solving problems without even explicitly knowing the rules of the task at hand reinforcement learning research progress has been the epicenter of many breakthrough ideas. In this paper, we analyze the methodological issues in deep reinforcement learning. We introduce the theoretical foundations of the underlying causes outlining that the asymptotic performance of deep reinforcement learning algorithms does not have a monotone relationship to the performance in the regimes where data becomes scarce. The extensive large-scale empirical analysis provided in our paper discovers that a major line of deep reinforcement learning research under the canonical methodological choices resulted in suboptimal conclusions.

## 1 INTRODUCTION

Founded on rigorous theoretical guarantees (Sutton, 1984; Watkins, 1989; Barto & Singh, 1990; Barto et al., 1995), reinforcement learning research achieved high acceleration upon the proposal of the initial study on approximating the state-action value function via deep neural networks (Mnih et al., 2015). Following this initial study a line of highly successful deep reinforcement learning algorithms have been proposed (Hasselt et al., 2016; Wang et al., 2016; Hessel et al., 2018; 2021; Kapturowski et al.; Zhu et al., 2024) from focusing on different architectural ideas to employing estimators targeting overestimation, all of which were designed and tested in the high-data regime (i.e. two hundred million frame training). An alternative recent line of research with an extensive amount of publications focused on pushing the performance bounds of deep reinforcement learning policies in the low-data regime, i.e. with one hundred thousand environment interaction training. Many different ideas in current reinforcement learning research, from model-based reinforcement learning to increasing sample efficiency with observation regularization, gained acceleration in several research directions based on policy performance comparisons demonstrated in the Arcade Learning Environment 100K benchmark.

In this paper, we focus on the canonical methodological choices made in deep reinforcement learning research and demonstrate that there is a significant overlooked underlying premise driving this line of research without being explicitly discussed: that the performance profiles of deep reinforcement learning algorithms have a monotonic relationship with different sample-complexity regimes. This implicit assumption, that is commonly shared amongst a large collection of low-data regime studies, shapes how the canonical methodological choices are made in deep reinforcement learning research and represents a prominent misdirection in scientific progress. The suboptimal conclusions obtained from these canonical choices shape future research directions with incorrect reasoning. These methodological decisions fueled by the incorrect conclusions, further influence the overall research efforts directed towards certain ideas for several years following. Thus, in our paper we target these underlying premises and aim to answer the following questions:

- *What are the canonical methodological choices made in deep reinforcement learning that fundamentally affects the conclusions made?*

- *What is the foundational relationship between sample complexity and the algorithmic performance from the data-scarce regime to the asymptotic regime?*

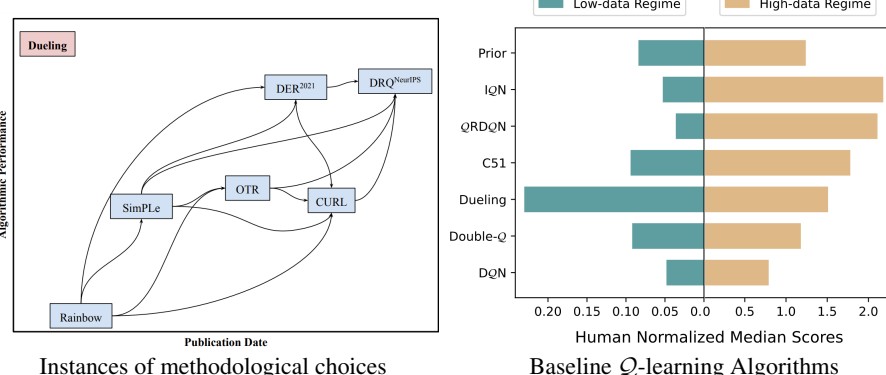

Figure 1: Left: Baseline comparisons visualization on the Arcade Learning Environment 100K benchmark. A directed arrow from Algorithm A to Algorithm B means that the algorithm B provides comparison against the algorithm A as a baseline. Right: Distributional vs baseline $\mathcal{Q}$ comparison of algorithms that were proposed and developed in the high-data regime in the Arcade Learning Environment in both high-data regime and low-data regime.

Hence, to be able to answer the questions raised above in our paper we focus on sample complexity in deep reinforcement learning and make the following contributions:

- We analyze the canonical methodological choices in deep reinforcement learning research, and introduce the theoretical foundations on how these methodological choices affect algorithm design, performance comparisons and algorithmic conclusions. Our analysis lays the foundations on the tight relationship between algorithmic performance and the sample complexity regimes.

- Our theoretical analysis proves that the performance profile has a non-monotonic relationship with the asymptotic sample complexity and the low-data sample complexity regime. Regarding the central focus of the large scale implicit assumption instances, our results reveal that the canonical methodological choices made in deep reinforcement learning research have led to incorrect conclusions.

- We conduct large scale extensive experiments for a comprehensive diverse portfolio of deep reinforcement learning baseline algorithms in both the low-data regime and the high-data regime Arcade Learning Environment benchmark. Our results demonstrate that recent algorithms proposed and evaluated in the Arcade Learning Environment 100K benchmark are significantly affected by the implicit assumption on the relationship between performance profiles and sample complexity.

## 2 BACKGROUND AND PRELIMINARIES

The reinforcement learning problem is formalized as a Markov Decision Process (MDP) represented as a tuple $\langle S, A, \mathcal{P}, \mathcal{R}, \gamma, \rho_0 \rangle$ where $S$ represents the state space, $A$ represents the set of actions, $\mathcal{P} : S \times A \to \Delta(S)$ represents the transition probability kernel that maps a state and an action pair to a distribution on states, $\mathcal{R} : S \times A \to \mathbb{R}$ represents the reward function, and $\gamma \in (0, 1]$ represents the discount factor. The aim in reinforcement learning is to learn an optimal policy $\pi(s, a)$ that outputs the probability of taking action $a$ in state $s$, $\pi : S \times A \to \mathbb{R}$ that will maximize expected cumulative discounted rewards $R = \mathbb{E}_{a_t \sim \pi(s_t, \cdot), s_{t+1} \sim \mathcal{P}(\cdot|s_t, a_t)} \sum_t \gamma^t \mathcal{R}(s_t, a_t, s_{t+1})$. This objective is achieved by constructing a state-action value function that learns for each state-action pair the expected cumulative discounted rewards that will be obtained if action $a \in A$ is executed in state $s \in S$ $\mathcal{Q}(s_t, a_t) = \mathcal{R}(s_t, a_t, s_{t+1}) + \gamma \sum_{s_t} \mathcal{P}(s_{t+1}|s_t, a_t) \mathcal{V}(s_{t+1})$. In settings where the state space and/or action space is large enough that the state-action value function $\mathcal{Q}(s, a)$ cannot be held in a tabular form, a function approximator is used. Thus, for deep reinforcement learning the $\mathcal{Q}$-function is approximated via deep neural networks $\theta_{t+1} = \theta_t + \alpha(\mathcal{R}(s_t, a_t, s_{t+1}) + \gamma \mathcal{Q}(s_{t+1}, \arg\max_a \mathcal{Q}(s_{t+1}, a; \theta_t); \theta_t) - \mathcal{Q}(s_t, a_t; \theta_t)) \nabla_{\theta_t} \mathcal{Q}(s_t, a_t; \theta_t)$.

**Dueling Architecture:** At the end of convolutional layers for a given deep $\mathcal{Q}$-Network, the dueling architecture outputs two streams of fully connected layers for both estimating the state values $\mathcal{V}(s)$

and the advantage $\mathcal{A}(s, a)$ for each action in a given state $s$, $\mathcal{A}(s, a) = \mathcal{Q}(s, a) - \max_a \mathcal{Q}(s, a)$. In particular, the last layer of the dueling architecture contains the forward mapping (Wang et al., 2016)

$$\mathcal{Q}(s, a; \theta, \alpha, \beta) = \mathcal{V}(s; \theta, \beta) + \left( \mathcal{A}(s, a; \theta, \alpha) - \max_{a' \in A} \mathcal{A}(s, a'; \theta, \alpha) \right)$$

where $\theta$ represents the parameters of the convolutional layers and $\alpha$ and $\beta$ represent the parameters of the fully connected layers outputting the advantage and state value estimates respectively.

**Learning the State-Action Value Distribution:** The initial algorithm proposed to learn the state-action value distribution (Agarwal et al., 2022) is C51. The projected Bellman update for the $i^{\text{th}}$ atom is computed as

$$(\Phi \mathcal{T} \mathcal{Z}_\theta(s_t, a_t))_i = \sum_{j}^{\mathcal{N}-1} \left[ 1 - \frac{|[\mathcal{T} z_j]_{v_{\min}}^{v_{\max}} - z_i|}{\Delta z} \right]_0^1 \tau_j(s_{t+1}, \max_{a \in A} \mathbb{E} \mathcal{Z}_\theta(s_{t+1}, a))$$

where $\mathcal{Z}_\theta(s_t, a_t)$ is the value distribution, $z_i = v_{\min} + i\Delta z : 0 \leq i < \mathcal{N}$ represents the set of atoms in categorical learning, and the atom probabilities are learnt as a parametric model

$$\tau_i(s_t, \max_{a \in A} \mathbb{E} \mathcal{Z}_\theta(s_t, a)) = \frac{e^{\theta_i(s_t, a_t)}}{\sum_j e^{\theta_j(s_t, a_t)}} \quad , \quad \Delta z := \frac{v_{\max} - v_{\min}}{\mathcal{N} - 1}$$

Following this baseline algorithm the $\mathcal{Q}$RD$\mathcal{Q}$N algorithm is proposed to learn the quantile projection of the state-action value distribution

$$\mathcal{T} \mathcal{Z}(s_t, a_t) = \mathcal{R}(s_t, a_t, s_{t+1}) + \gamma \mathcal{Z}(s_{t+1}, \arg\max_{a \in A} \mathbb{E}_{z \sim \mathcal{Z}(s_{t+1}, a_{t+1})}[z])$$

with $s_{t+1} \sim \mathcal{P}(\cdot | s_t, a_t)$ where $\mathcal{Z} \in Z$ represents the quantile distribution of an arbitrary value function. Following this study the I$\mathcal{Q}$N algorithm is proposed (i.e. implicit quantile networks) to learn the full quantile function instead of learning a discrete set of quantiles as in the $\mathcal{Q}$RD$\mathcal{Q}$N algorithm. The I$\mathcal{Q}$N algorithm objective is to minimize the loss function

$$\mathcal{L} = \frac{1}{\mathcal{K}} \sum_{i=1}^{\mathcal{K}} \sum_{j=1}^{\mathcal{K}'} \rho_{\delta_i}(\mathcal{R}(s_t, a_t, s_{t+1}) + \gamma \mathcal{Z}_{\delta_{j'}}(s_{t+1}, \arg\max_{a \in A} \mathcal{Q}_\beta(s_{t+1}, a_{t+1})) - \mathcal{Z}_{\delta_i}(s_t, a_t))$$

where $\rho_\delta$ represents the Huber quantile regression loss, and $\mathcal{Q}_\beta = \int_0^1 \mathcal{F}_{\mathcal{Z}}^{-1}(\delta) d\beta(\delta)$. Note that $\mathcal{Z}_\delta = \mathcal{F}_{\mathcal{Z}}^{-1}(\delta)$ is the quantile function of the random variable $\mathcal{Z}$ at $\delta \in [0, 1]$.

# 3 LOW-DATA REGIME VERSUS ASYMPTOTIC PERFORMANCE

Our paper discovers both with extensive and comprehensive empirical analysis and theoretical investigation that asymptotic performance of reinforcement learning algorithms does not necessarily provide any information nor indication on their relative performance ranking in the low-data regime. The results provided in Section 5 extensively demonstrate that a large body of work in reinforcement learning research carried this assumption and resulted in incorrect conclusions. In this section we introduce the foundational basis for this discovery revealed by our empirical analysis provided in Section 5 in optimization of non-stationary policies, i.e. rewards and transitions can vary with each step in an episode, in undiscounted, finite-horizon MDPs with linear function approximation. In particular, a finite horizon MDP is represented as a tuple $\langle S, A, \mathcal{P}, \mathcal{R}, \mathcal{H} \rangle$ where $S$ is the set of states, and $A$ represents the set of actions. For each timestep $t \in [\mathcal{H}] = \{1, \ldots, \mathcal{H}\}$, state $s$, and action $a$ the transition probability kernel $\mathcal{P}_t(s_{t+1} | s_t, a_t)$ gives the probability distribution over the next state, and the reward $\mathcal{R}_t(s_t, a_t, s_{t+1})$ gives the immediate rewards. A non-stationary policy $\pi = (\pi_1, \ldots, \pi_{\mathcal{H}})$ induces a state-action value function given by

$$\mathcal{Q}_t^\pi(s_t, a_t) = \mathcal{R}_t(s_t, a_t, s_{t+1}) + \mathbb{E}_{\substack{s_t \sim \mathcal{P}_t(s_{t+1} | s_t, a_t) \\ a_t \sim \pi}} \left[ \sum_{h=t+1}^{\mathcal{H}} \mathcal{R}_t(s_h, \pi_h(s_h), s_{h+1}) \middle| s_t, a_t \right]$$

where we let $\pi(s)$ be the action taken by the policy $\pi$ in state $s$, and the corresponding value function $\mathcal{V}_t^\pi(s_t) = \mathcal{Q}_t(s_t, \pi(s_t))$. The optimal non-stationary policy $\pi^*$ has value function $\mathcal{V}_t^*(s_t) = \mathcal{V}_t^{\pi^*}(s_t)$

satisfying $\mathcal{V}_t^*(s_t) = \sup_\pi \mathcal{V}_t^\pi(s_t)$. The objective is to learn a sequence of non-stationary policies $\pi^k$ for $k \in \{1, \ldots, \mathcal{K}\}$ while interacting with an unknown MDP in order to minimize the regret, which is measured asymptotically over $\mathcal{K}$ episodes of length $\mathcal{H}$

$$\text{REGRET}(\mathcal{K}) = \sum_{k=1}^{\mathcal{K}} \left( \mathcal{V}_1^*(s_1^k) - \mathcal{V}_1^{\pi^k}(s_1^k) \right) \tag{1}$$

where $s_1^k \in S$ is the starting state of the $k$-th episode. Regret sums up the gap between the expected rewards obtained by the sequence of learned policies $\pi^k$ and those obtained by $\pi^*$ when learning for $\mathcal{K}$ episodes. In the linear function approximation setting there is a feature map $\phi_t : S \times A \to \mathbb{R}^{d_t}$ for each $t \in [\mathcal{H}]$ that sends a state-action pair $(s, a)$ to the $d_t$-dimensional vector $\phi_t(s, a)$. Then, the state-action value function $\mathcal{Q}_t(s_t, a_t)$ is parameterized by a vector $\theta_t \in \mathbb{R}^{d_t}$ so that $\mathcal{Q}_t(\theta_t)(s_t, a_t) = \phi_t(s, a)^\top \theta_t$. Recent theoretical work in this setting gives an algorithm along with a lower bound that matches the regret achieved by the algorithm up to logarithmic factors.

**Theorem 3.1** (Zanette et al. (2020)). *Under appropriate normalization assumptions there is an algorithm that learns a sequence of policies $\pi^k$ achieving regret $\text{REGRET}(\mathcal{K}) = \tilde{O}\left( \sum_{t=1}^{\mathcal{H}} d_t \sqrt{\mathcal{K}} + \sum_{t=1}^{\mathcal{H}} \sqrt{d_t} \mathcal{I} \mathcal{K} \right)$, where $\mathcal{I}$ is the inherent Bellman error. Furthermore, this regret bound is optimal for this setting up to logarithmic factors in $d_t, \mathcal{K}$ and $\mathcal{H}$ whenever $\mathcal{K} = \Omega((\sum_{t=1}^{\mathcal{H}} d_t)^2)$, in the sense that for any level of inherent Bellman error $\mathcal{I}$ and sequence of feature dimensions $\{d_t\}_{t=1}^H$, there exists a class of MDPs $\mathcal{C}(\mathcal{I}, \{d_t\}_{t=1}^H)$ where any algorithm achieves at least as much regret on at least one MDP in the class.*

The class of MDPs $\mathcal{C}(\mathcal{I}, \{d_t\}_{t=1}^H)$ constructed in Theorem 3.1 additionally satisfies the following properties. First, every MDP in $\cup_{\mathcal{I}, \{d_t\}_{t=1}^H} \mathcal{C}(\mathcal{I}, \{d_t\}_{t=1}^H)$ has the same transitions (up to renaming of states and actions). Second, for each fixed value of the inherent Bellman error $\mathcal{I}$ and the dimensions $\{d_t\}_{t=1}^H$, every MDP in $\mathcal{C}(\mathcal{I}, \{d_t\}_{t=1}^H)$ utilizes the same feature map $\phi_t(s_t, a_t)$. Thus one can view the class $\mathcal{C}(\mathcal{I}, \{d_t\}_{t=1}^H)$ as encoding one "underlying" true environment (defined by the transitions), with varying values of $\mathcal{I}$ and $\{d_t\}_{t=1}^H$ corresponding to varying levels of function approximation accuracy, and model capacity for the underlying environment. For simplicity of notation we will focus on the setting where $d_t = d$ for all $t \in \{1, \ldots H\}$ and write $\mathcal{C}(\mathcal{I}, d)$ for the class of MDPs constructed in Theorem 3.1 for this setting. Utilizing this point of view, we can then prove the following theorem on the relationship between the performance in the asymptotic and low-data regimes.

**Theorem 3.2** (Non-monotonocity Across Regimes). *For any $\epsilon > 0$, let $d_\alpha$ be any feature dimension, and let $d_\beta = d_\alpha^{1-\epsilon/2}$. Then there exist thresholds $\mathcal{K}_{low} < \mathcal{K}_{high}$ and inherent Bellman error levels $\mathcal{I}_\beta > \mathcal{I}_\alpha$ such that*

1. *There is an algorithm achieving regret $\text{REGRET}_{low}(\mathcal{K})$ when $\mathcal{K} < \mathcal{K}_{low}$ for all MDPs in $\mathcal{C}(\mathcal{I}_\beta, d_\beta)$. However, every algorithm has regret at least $\widetilde{\Omega}\left( d_\beta^{\epsilon/2} \text{REGRET}_{low}(\mathcal{K}) \right)$ when $\mathcal{K} < \mathcal{K}_{low}$ on some MDP $M \in \mathcal{C}(\mathcal{I}_\alpha, d_\alpha)$.*

2. *There is an algorithm achieving regret $\text{REGRET}_{high}(\mathcal{K})$ when $\mathcal{K} > \mathcal{K}_{high}$ for all MDPs in $\mathcal{C}(\mathcal{I}_\alpha, d_\alpha)$. However, every algorithm has regret at least $\widetilde{\Omega}(d_\alpha^\epsilon \text{REGRET}_{high}(\mathcal{K}))$ on some MDP $M \in \mathcal{C}(\mathcal{I}_\beta, d_\beta)$ when $\mathcal{K} > \mathcal{K}_{high}$.*

*Proof.* Let $\epsilon > 0$ and consider $d_\beta = d_\alpha^{1-\frac{\epsilon}{2}}, \mathcal{I}_\beta = \frac{1}{d_\alpha^\epsilon \sqrt{d_\beta}}, \mathcal{I}_\alpha = \frac{1}{d_\alpha^{\frac{1}{2}+2\epsilon}}, \mathcal{K}_{low} = d_\alpha^{2+\epsilon}, \mathcal{K}_{high} = d_\alpha^{2+4\epsilon}$

We begin with the proof of part 1. Therefore, for $\mathcal{K} < \mathcal{K}_{low}, \sqrt{d_\beta} \mathcal{I}_\beta \mathcal{K} = d_\alpha^{-\epsilon} \mathcal{K} < d_\alpha^{1-\frac{\epsilon}{2}} \sqrt{\mathcal{K}} = d_\beta \sqrt{\mathcal{K}}$. Therefore, by Theorem 3.1 there exists an algorithm achieving regret

$$\text{REGRET}_{low}(\mathcal{K}) = \tilde{O}\left( \mathcal{H} d_\beta \sqrt{\mathcal{K}} + \mathcal{H} \sqrt{d_\beta} \mathcal{I}_\beta \mathcal{K} \right) = \widetilde{O}\left( d_\beta \sqrt{\mathcal{K}} \right)$$

in every MDP $M \in \mathcal{C}(\mathcal{I}_\beta, d_\beta)$. Further, since $\mathcal{K}_{low} = d_\alpha^{2+\epsilon} > \widetilde{\Omega}\left(d_\alpha^2\right)$, the lower bound from Theorem 3.1 applies to the class of MDPs $\mathcal{C}(\mathcal{I}_\alpha, d_\alpha)$ for all $\mathcal{K} \in \left[ \widetilde{\Omega}\left(d_\alpha^2\right), \mathcal{K}_{low} \right]$. In particular, every algorithm receives regret at least

$$\text{REGRET}(\mathcal{K}) = \widetilde{\Omega}\left( \mathcal{H} d_\alpha \sqrt{\mathcal{K}} + \mathcal{H} \sqrt{d_\alpha} \mathcal{I}_\alpha \mathcal{K} \right) > \widetilde{\Omega}\left( \mathcal{H} d_\beta^{\frac{1}{1-\epsilon/2}} \sqrt{\mathcal{K}} \right) > \widetilde{\Omega}\left( \mathcal{H} d_\beta^{\frac{\epsilon/2}{1-\epsilon/2}} d_\beta \sqrt{\mathcal{K}} \right)$$

Thus, $\text{REGRET}(\mathcal{K}) > \widetilde{\Omega}\left(d_\beta^{\epsilon/2}\text{REGRET}_{\text{low}}(\mathcal{K})\right)$. For part 2, note that for $\mathcal{K} > \mathcal{K}_{\text{high}}$ we have both $\sqrt{d_\alpha}\mathcal{I}_\alpha\mathcal{K} = d_\alpha^{-2\epsilon}\mathcal{K} > d_\alpha^{-2\epsilon}\sqrt{\mathcal{K}\cdot\mathcal{K}_{\text{high}}} > d_\alpha\sqrt{\mathcal{K}}$ and $\sqrt{d_\beta}\mathcal{I}_\beta\mathcal{K} > d_\alpha^{-\epsilon}\sqrt{\mathcal{K}\cdot\mathcal{K}_{\text{low}}} = d_\alpha^{1+\epsilon}\sqrt{\mathcal{K}} > d_\beta\sqrt{\mathcal{K}}$. Therefore by Theorem 3.1 that for $\mathcal{K} > \mathcal{K}_{\text{low}}$ there exists an algorithm achieving regret

$$\text{REGRET}_{\text{high}}(\mathcal{K}) = \tilde{O}\left(\mathcal{H}d_\alpha\sqrt{\mathcal{K}} + \mathcal{H}\sqrt{d_\alpha}\mathcal{I}_\alpha\mathcal{K}\right) = \tilde{O}\left(\mathcal{H}\sqrt{d_\alpha}\mathcal{I}_\alpha\mathcal{K}\right).$$

for every MDP $M \in \mathcal{C}(\mathcal{I}_\alpha, d_\alpha)$. However, by the lower bound in Theorem 3.1, for $\mathcal{K} > \mathcal{K}_{\text{low}}$ every algorithm receives regret at least

$$\text{REGRET}(\mathcal{K}) = \widetilde{\Omega}\left(\mathcal{H}d_\beta\sqrt{\mathcal{K}} + \mathcal{H}\sqrt{d_\beta}\mathcal{I}_\beta\mathcal{K}\right) > \widetilde{\Omega}\left(\mathcal{H}\sqrt{d_\beta}\mathcal{I}_\beta\mathcal{K}\right) = \widetilde{\Omega}\left(\mathcal{H}d_\alpha^{-\epsilon}\mathcal{K}\right)$$

$$= \widetilde{\Omega}\left(d_\alpha^\epsilon\mathcal{H}d_\alpha^{-2\epsilon}\mathcal{K}\right) = \widetilde{\Omega}\left(d_\alpha^\epsilon\mathcal{H}\sqrt{d_\alpha}\mathcal{I}_\alpha\mathcal{K}\right) > \widetilde{\Omega}\left(d_\alpha^\epsilon\text{REGRET}_{\text{high}}(\mathcal{K})\right) \qquad \square$$

Theorem 3.2 introduces the provable trade-off between performance in the low-data regime (i.e. $\mathcal{K} < \mathcal{K}_{\text{low}}$) and the high-data regime (i.e. $\mathcal{K} > \mathcal{K}_{\text{high}}$). In particular, in the low-data regime lower capacity function approximation, i.e. lower feature dimension $d_\beta$, with larger approximation error, i.e. larger inherent Bellman error $\mathcal{I}_\beta$, can provably outperform larger capacity models, i.e. feature dimension $d_\alpha$, with smaller approximation error, i.e. inherent Bellman error $\mathcal{I}_\alpha$. Furthermore, the relative performance is reversed in the high-data regime $\mathcal{K} > \mathcal{K}_{\text{high}}$. Thus, asymptotic performance of an algorithm is neither indicative nor carries any relevant information on the expected performance of the algorithm when training data is scarce (i.e. limited).

# 4 LOWER BOUNDS FOR LEARNING THE STATE-ACTION VALUE DISTRIBUTION

The instances of the implicit assumption that the performance profile of an algorithm in the high-data regime will translate to the low-data regime monotonically appear in almost all of the studies conducted in the low-data regime. In particular, we see that when this line of work was being conducted the best performing algorithm in the high-data regime was based on learning the state action value distribution. Hence, there are many cases in the literature (e.g. DRQ, OTR, DER, CURL, SimPLE, Efficient-Zero) where all the newly proposed algorithms in the low-data regime are being compared to an algorithm that learns the state-action value distribution, under the implicit assumption that the algorithm that learns the state-action value distribution must achieve the current best performance in the low-data regime. The large scale experiments provided in Section 5 demonstrate the impact of this implicit assumption in the low-data regime deep reinforcement learning algorithm design. In particular, the results reported in Section 5 prove that the performance profile of an algorithm in the high-data regime does not monotonically transfer to the low-data regime. Due to this extensive focus throughout the literature on low-data regime comparisons to algorithms that learn the state action value distribution, we provide additional theoretical justification for the empirically observed sample complexity results in the low to high-data regime in deep reinforcement learning.

To obtain theoretical insight into the larger sample complexity exhibited by learning the state-action value distribution we consider the fundamental comparison between learning the distribution of a random variable $\mathcal{X}$ versus only learning the mean $\mathbb{E}[\mathcal{X}]$. In the base algorithm that learns the state-action value distribution the goal is to learn a distribution over state-action values that has finite support. It is well-known that learning a discrete distribution to error $\epsilon$ in total variation distance requires more samples than estimating the mean to within error epsilon (see Proposition B.1). Although this fact implies that learning the state-action value distribution has an intrinsically higher sample complexity than that of standard $\mathcal{Q}$-learning, it does not provide insights into the comparison of an error of $\epsilon$ in the mean with an error of $\epsilon$ in total variation distance. Hence, the following proposition demonstrates a precise justification of the comparison: whenever there are two different actions where the true mean state-action values are within $\epsilon$, an approximation error of $\epsilon$ in total variation distance for the state-action value distribution of one of the actions can be sufficient to reverse the order of the means.

**Proposition 4.1** (*Accuracy of Mean vs Distribution*)**.** *Fix a state $s$ and consider two actions $a, \hat{a}$. Let $\mathcal{X}(s, a)$ be the random variable distributed as the true state-action value distribution of $(s, a)$, and $\mathcal{X}(s, \hat{a})$ be the same for $(s, \hat{a})$. Suppose that $\mathbb{E}[\mathcal{X}(s, a)] = \mathbb{E}[\mathcal{X}(s, \hat{a})] + \epsilon$. Then there is a random variable $\mathcal{Y}$ such that $d_{TV}(\mathcal{Y}, \mathcal{X}(s, a)) \leq \epsilon$ and $\mathbb{E}[\mathcal{X}(s, \hat{a})] \geq \mathbb{E}[\mathcal{Y}]$.*

*Proof.* Let $\tau^* \in \mathbb{R}$ be the infimum $\tau^* = \inf\{\tau \in \mathbb{R} \mid \mathbb{P}[\mathcal{X}(s,a) \geq \tau] = \epsilon\}$ i.e. $\tau^*$ is the first point in $\mathbb{R}$ such that $\mathcal{X}(s,a)$ takes values at least $\tau^*$ with probability exactly $\epsilon$. Next let the random variable $\mathcal{Y}$ be defined by the following process. First, sample the random variable $\mathcal{X}(s,a)$. If $\mathcal{X}(s,a) \geq \tau^*$, then output $\tau^* - 1$. Otherwise, output the sampled value of $\mathcal{X}(s,a)$. Observe that the probability distributions of $\mathcal{Y}$ and $\mathcal{X}(s,a)$ are identical except at the point $\tau^* - 1$ and on the interval $[\tau^*, \infty)$. Let $\lambda$ be the Lebesgue measure on $\mathbb{R}$. By construction of $\mathcal{Y}$ the total variation distance is given by

$$d_{TV}(\mathcal{Y}, \mathcal{X}) = \frac{1}{2} \int_{\mathbb{R}} \big| \mathbb{P}[\mathcal{X}(s,a) = z] - \mathbb{P}[\mathcal{Y} = z] \big| \, d\lambda(z) = \frac{1}{2} \big| \mathbb{P}[\mathcal{X}(s,a) = \tau^* - 1]$$

$$- \mathbb{P}[\mathcal{Y} = \tau^* - 1] \big| + \frac{1}{2} \int_{[\tau^*, \infty)} \big| \mathbb{P}[\mathcal{X}(s,a) = z] - \mathbb{P}[\mathcal{Y} = z] \big| \, d\lambda(z) = \frac{\epsilon}{2} + \frac{\epsilon}{2} = \epsilon.$$

Next note that the expectation of $\mathcal{Y}$ is given by

$$\mathbb{E}[\mathcal{Y}] = \epsilon(\tau^* - 1) + \int_{(-\infty, \tau^*]} z \mathbb{P}[\mathcal{X}(s,a) = z] \, d\lambda(z) = \epsilon(\tau^* - 1) + \int_{\mathbb{R}} z \mathbb{P}[\mathcal{X}(s,a) = z] \, d\lambda(z)$$

$$- \int_{(\tau^*, \infty]} z \mathbb{P}[\mathcal{X}(s,a) = z] \, d\lambda(z) \leq \epsilon(\tau^* - 1) + \mathbb{E}[\mathcal{X}(s,a)] - \epsilon\tau^* = \mathbb{E}[\mathcal{X}(s,a)] - \epsilon$$

where the inequality follows from the fact that $\mathcal{X}$ takes values larger than $\tau^*$ with probability $\epsilon$. $\qquad\square$

Proposition 4.1 shows that, in the case where the mean state-action values are within $\epsilon$, unless the state-action value distribution is learned to within total-variation distance $\epsilon$, the incorrect action may be selected by the policy that learns the state-action value distribution. Therefore, it is natural to compare the sample complexity of learning the state-action value distribution to within total-variation distance $\epsilon$ with the sample complexity of simply learning the mean to within error $\epsilon$, as is done in Proposition B.1.

## 4.1 Learning State-Action Values with Unknown Support

The setting considered in Proposition B.1 most readily applies to the base algorithm that learns the state-action value distribution C51, which attempts to directly learn a discrete distribution with known support in order to approximate the state-action value distribution. However, further advances in learning the state-action value distribution including $\mathcal{Q}$RD$\mathcal{Q}$N and I$\mathcal{Q}$N do away with the assumption that the support of the distribution is known. This allows a more flexible representation in order to more accurately represent the true distribution on state-action values, but, as we will show, potentially leads to a further increase in the sample complexity. The $\mathcal{Q}$RD$\mathcal{Q}$N algorithm models the distribution of state-action values as a uniform mixture of $\mathcal{N}$ Dirac deltas on the reals i.e. $\mathcal{Z}(s,a) = \frac{1}{\mathcal{N}} \sum_{i=1}^{\mathcal{N}} \delta_{\theta_i(s,a)}$, where $\theta_i(s,a) \in \mathbb{R}$ is a parametric model.

**Proposition 4.2** (*Sample Complexty with Unknown Support*). *Let $\mathcal{N} > \mathcal{M} \geq 2$, $\epsilon > \frac{\mathcal{M}}{4\mathcal{N}}$, and $\theta_i \in \mathbb{R}$ for $i \in [\mathcal{N}]$. The number of samples required to learn a distribution of the form $\mathcal{Z} = \frac{1}{\mathcal{N}} \sum_{i=1}^{\mathcal{N}} \delta_{\theta_i}$ to within total variation distance $\epsilon$ is $\Omega\left(\frac{\mathcal{M}}{\epsilon^2}\right)$.*

The proof is provided in the appendix. Depending on the tolerance to the approximation error, the lower bound in Proposition 4.2 can be significantly larger than that of Proposition B.1. For example if the desired approximation error is $\epsilon = \frac{1}{8}$ one can take $\mathcal{M} = \frac{\mathcal{N}}{2}$. In this case if the value of $k$ in Proposition B.1 satisfies $k = o(\mathcal{N})$, then the sample complexity in Proposition 4.2 is asymptotically larger than that of Proposition B.1.

## 5 Large Scale Empirical Analysis

The empirical analysis is conducted in the Arcade Learning Environment (ALE) (Bellemare et al., 2013). The Double $\mathcal{Q}$-learning algorithm is trained via Double Deep $\mathcal{Q}$-Network (Hasselt et al., 2016) initially proposed by van Hasselt (2010). The dueling algorithm is trained via Wang et al. (2016). The policies that learn the state-action value distribution are trained via the C51 algorithm (Bellemare et al., 2017), I$\mathcal{Q}$N (Dabney et al., 2018a) and $\mathcal{Q}$RD$\mathcal{Q}$N (Dabney et al., 2018b). To provide a complete picture of the sample complexity we conducted our experiments in both low-data, i.e.

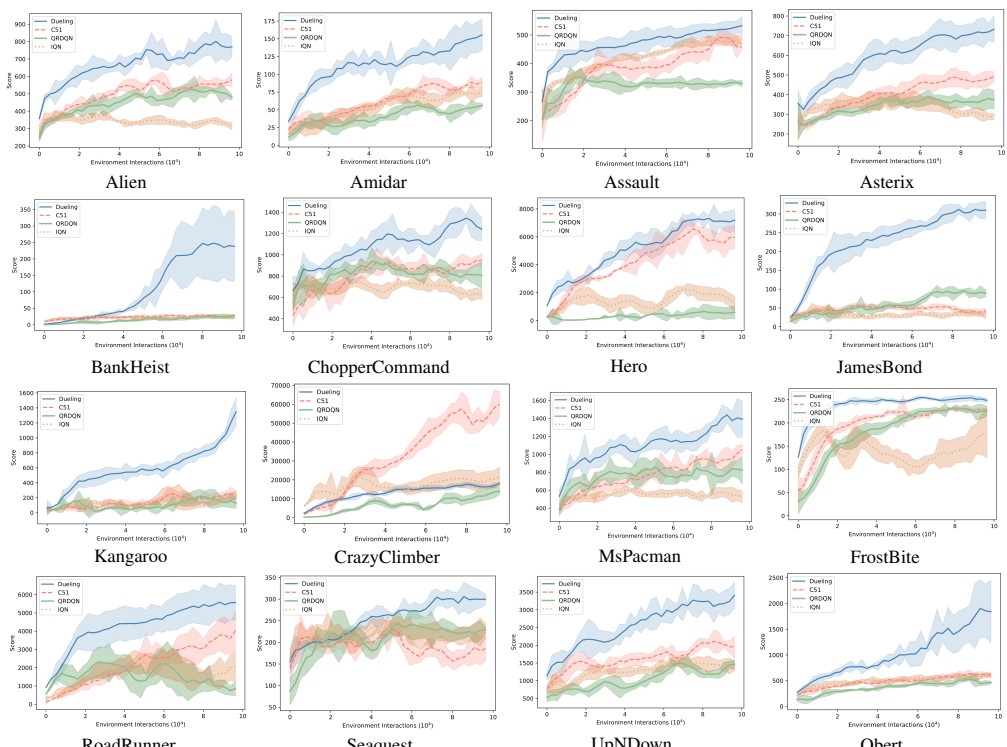

Figure 2: The learning curves of Alien, Amidar, Asterix, BankHeist, ChopperCommand, Hero, CrazyClimber, JamesBond, Kangaroo, MsPacman, FrostBite, Qbert, RoadRunner, Seaquest and UpNDown with dueling architecture, C51, I$\mathcal{Q}$N and $\mathcal{Q}$RD$\mathcal{Q}$N algorithms in the Arcade Learning Environment with 100K environment interaction training.

the Arcade Learning Environment 100K benchmark, and high data regime, i.e. baseline 200 million frame training. The prior algorithm refers to the prioritized experience replay algorithm proposed by Schaul et al. (2016). All of the results are reported with the standard error of the mean in all of the tables and figures in the paper. The experiments are run with JAX (Bradbury et al., 2018), with Haiku as the neural network library, Optax (Hessel et al., 2020) as the optimization library, and RLax for the reinforcement learning library (Babuschkin et al., 2020). More details on the hyperparameters and direct references to the implementations can be found in the supplementary material. The line of algorithms that are closely examined in the low data regime includes DRQ (Yarats et al., 2021), OTR (Kielak, 2019), DER (van Hasselt et al., 2019), CuRL (Laskin et al., 2020), SimPLe (Kaiser et al., 2020). A line of instances of the implicit assumption that our paper discovers, and provides an explicit analysis for, appears from the start of the creation of the ALE 100K benchmark and extends to many recent studies (Kaiser et al., 2020; Laskin et al., 2020; Kielak, 2019; Yarats et al., 2021; Schwarzer et al., 2023). Note that human normalized score is computed as follows: $\text{Score}_{\text{HN}} = (\text{Score}_{agent} - \text{Score}_{random})/(\text{Score}_{human} - \text{Score}_{random})$.

**Implicit Assumptions on Monotonocity Cause Suboptimal and Incorrect Conclusions:** Our extensive large-scale empirical analysis demonstrates that a major line of research conducted in the past five years resulted in suboptimal conclusions. In particular, Figure 5 shows that a simple baseline dueling algorithm from 2016, by a canonical methodological choice was never included in the comparison benchmark, due to the implicit assumption that appears in all of the recent line of research that we have discussed in detail in Section 3 and presented in Figure 1. Thus, we demonstrate that this baseline algorithm in fact performs much better than many recent algorithms that claimed to be better than the baselines, even including algorithms that are built on top of the dueling algorithm.

Figure 2 reports learning curves for the I$\mathcal{Q}$N, $\mathcal{Q}$RD$\mathcal{Q}$N, dueling architecture and C51 for every MDP in the Arcade Learning Environment low-data regime 100K benchmark. These results demonstrate that the simple base algorithm dueling performs significantly better than any algorithm that focuses on learning the distribution when the training samples are limited. For a fair, direct and transparent

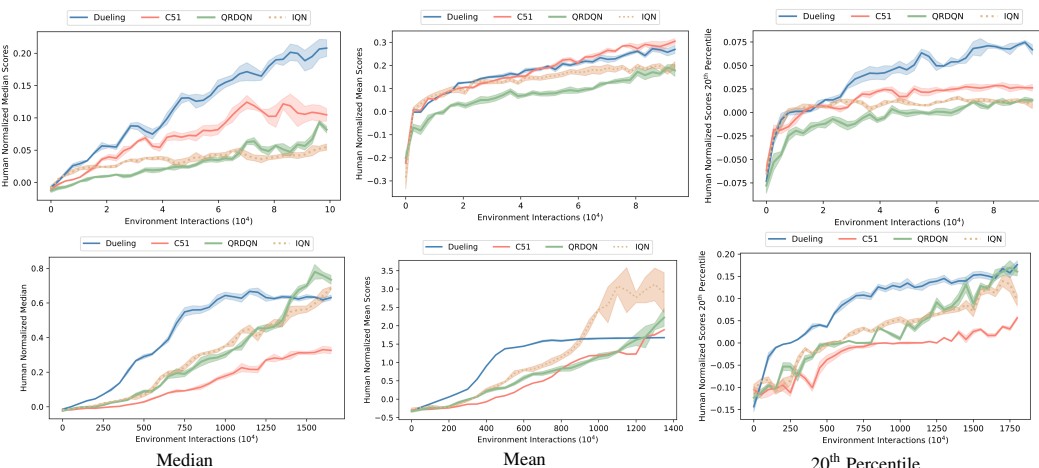

Figure 3: Up: Human normalized median, mean and 20$^{\text{th}}$ percentile results for the dueling algorithm, C51, I$\mathcal{Q}$N and $\mathcal{Q}$RD$\mathcal{Q}$N in the Arcade Learning Environment 100K benchmark. Down: Human normalized median, mean, and 20$^{\text{th}}$ percentile results for the dueling algorithm, C51, I$\mathcal{Q}$N and $\mathcal{Q}$RD$\mathcal{Q}$N in the high-data regime towards 200 million frame.

Table 1: Large scale comparison of $\mathcal{Q}$-based deep reinforcement learning algorithms with human normalized mean, median and 20$^{\text{th}}$ percentile results in the Arcade Learning Environment 100K benchmark for D$\mathcal{Q}$N (Mnih et al., 2015), deep Double-$\mathcal{Q}$ learning (Hasselt et al., 2016), dueling architecture (Wang et al., 2016), Prior (Schaul et al., 2016), C51, $\mathcal{Q}$RD$\mathcal{Q}$N and I$\mathcal{Q}$N.

| Algorithms | Human Normalized Median | Human Normalized Mean | 20$^{\text{th}}$ Percentile |
|---|---|---|---|
| D$\mathcal{Q}$N | 0.0481±0.0036 | 0.1535±0.0119 | 0.0031±0.0032 |
| Double-$\mathcal{Q}$ | 0.0920±0.0181 | **0.3169±0.0196** | 0.0341±0.0042 |
| Dueling | **0.2304±0.0061** | 0.2923±0.0060 | **0.0764±0.0037** |
| C51 | 0.0941±0.0081 | 0.3106±0.0199 | 0.0274±0.0024 |
| $\mathcal{Q}$RD$\mathcal{Q}$N | 0.0820±0.0037 | 0.2171±0.0098 | 0.0189±0.0031 |
| I$\mathcal{Q}$N | 0.0528±0.0058 | 0.2050±0.0123 | 0.0091±0.0011 |
| Prior | 0.0840±0.0018 | 0.2792±0.0123 | 0.0267±0.0042 |

comparison we kept the hyperparameters for the baseline algorithms in the low-data regime exactly the same with the DR$\mathcal{Q}^{\text{ICLR}}$ paper (see supplementary material for the full list and high-data regime hyperparameter settings). Note that the DR$\mathcal{Q}$ algorithm uses the dueling architecture without any distributional reinforcement learning. One intriguing takeaway from the results provided in Table 1 and the Figure 5[1] is the fact that the simple base algorithm dueling performs 15% better than the DR$\mathcal{Q}^{\text{NeurIPS}}$ implementation, and 11% less than the DR$\mathcal{Q}^{\text{ICLR}}$ implementation. Note that the original paper of the DR$\mathcal{Q}^{\text{ICLR}}$ algorithm provides comparison only to data-efficient Rainbow (DER) (van Hasselt et al., 2019) which inherently learns the state-action value distribution. The fact that the original paper that proposed data augmentation for deep reinforcement learning (i.e. DR$\mathcal{Q}^{\text{ICLR}}$) on top of the dueling architecture did not provide comparisons with the pure simple base dueling architecture (Wang et al., 2016) resulted in inflated performance profiles for the DR$\mathcal{Q}^{\text{ICLR}}$ algorithm.

More intriguingly, the comparisons provided in the DR$\mathcal{Q}^{\text{ICLR}}$ paper to the DER and OTR algorithms report the performance gained by DR$\mathcal{Q}^{\text{ICLR}}$ over DER is 82% and over OTR is 35%. However, if a direct comparison is made to the simple dueling algorithm as Table 1 demonstrates with the exact hyperparameters used as in the DR$\mathcal{Q}^{\text{ICLR}}$ paper the performance gain is utterly restricted to **11%**. Moreover, when it is compared to the reproduced results of DR$\mathcal{Q}^{\text{NeurIPS}}$ our results reveal that in fact there is a performance decrease due to utilizing the DR$\mathcal{Q}$ algorithm over dueling architecture. Thus, while our paper introduces the foundations on the non-monotonicity of the performance profiles

---

[1] DER$^{2021}$ refers to the re-implementation with random seed variations of the original paper data-efficient Rainbow (i.e. DER$^{2019}$) by van Hasselt et al. (2019). OTR refers to further implementation of the Rainbow algorithm by Kielak (2019). DR$\mathcal{Q}^{\text{NeurIPS}}$ refers to the re-implementation of the original DR$\mathcal{Q}$ algorithm published in ICLR as a spotlight presentation with the goal of achieving reproducibility with variation on the number of random seeds (Agarwal et al., 2021).

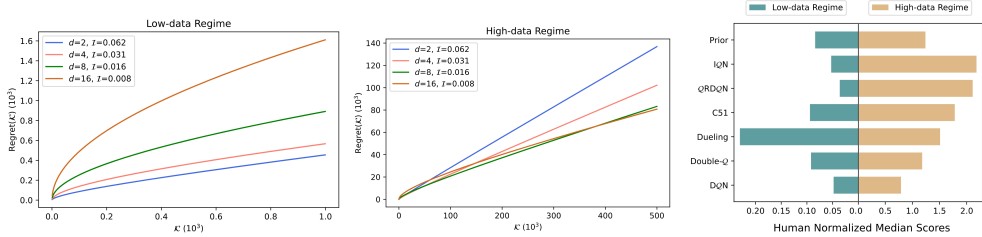

Regret in the Low-data Regime     Regret in the High-data Regime     Distributional vs Baseline $\mathcal{Q}$

Figure 4: Left: Regret in the low-data regime. Center: Regret in the high-data regime. Right: Distributional vs baseline $\mathcal{Q}$ comparison of algorithms that were proposed and developed in the high-data regime in the Arcade Learning Environment in both high-data regime and low-data regime.

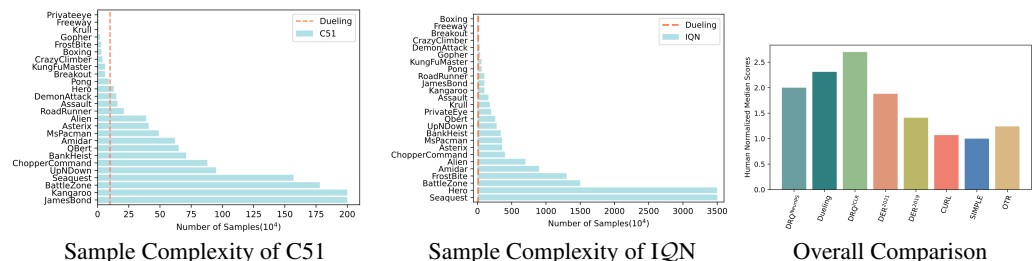

Sample Complexity of C51     Sample Complexity of I$\mathcal{Q}$N     Overall Comparison

Figure 5: Left: Number of samples, i.e. environment interactions, required by the base algorithm that learns the state-action value distribution to achieve the performance level achieved by the dueling algorithm. Center: Number of samples required by I$\mathcal{Q}$N to achieve the performance level achieved by dueling. Right: Overall comparison of algorithms recently developed in the low-data regime ALE 100K benchmark to the dueling algorithm that was designed in the high-data region.

from large-data regime to low-data regime, it further provides the basis on how we can compare algorithms with scientific rigor, and can influence future research to have more concrete and accurate performance profiles for algorithm development in both low-data and high-data regime.

**Providing Direct Comparison to Core Algorithms:** Algorithms that are built on top of a core reinforcement learning algorithm must provide a direct comparison to the algorithm they are built on top of. The case of DR$\mathcal{Q}$ demonstrates the significance of the direct comparison to the core algorithm. As our paper discovers and describes extensively, the monotonicity assumption on the performance ranking across regimes led a line of work to benchmark against algorithms in the low-data regime, assuming an algorithm that has the highest performance in the high-data regime must have the top ranked performance in the low-data regime. However, the results reported in Figure 1 and Figure 5 demonstrate that these implicit assumptions led to incorrect and suboptimal conclusions.

**Non-Monotonicity of Performance Ranking Across Regimes:** Table 1 reports the human normalized median, human normalized mean, and human normalized 20th percentile results over all of the MDPs from the 100K Arcade Learning Environment benchmark for D$\mathcal{Q}$N, Double-$\mathcal{Q}$, dueling, C51, $\mathcal{Q}$RD$\mathcal{Q}$N, I$\mathcal{Q}$N and prior. One important takeaway from the results reported in the Table 1 is the fact that one particular algorithm performance profile in 200 million frame training will not directly transfer to the low-data region. Figure 3 reports the learning curves of human normalized median, human normalized mean and human normalized 20th percentile for the dueling algorithm, C51, $\mathcal{Q}$RD$\mathcal{Q}$N, and I$\mathcal{Q}$N in the low-data region. These results once more demonstrate that the performance profile of the simple base algorithm dueling is significantly better than any other core algorithm that learns the state-action value distribution when the number of environment interactions are limited.

**Theoretical Analysis and the Inherent Bellman Error vs Dimensionality:** The left and center plots of Figure 4 report regret curves corresponding to the theoretical analysis in Theorem 3.2 for various choices of the feature dimensionality $d$ and the inherent Bellman error $\mathcal{I}$. In particular, the left plot shows the low-data regime where the number of episodes $\mathcal{K} < 1000$, while the right plot shows the high-data regime where $\mathcal{K}$ is as large as 500000. Notably, the relative ordering of the regret across the different choices of $d$ and $\mathcal{I}$ is completely reversed in the high-data regime when compared to the low-data regime. Recall from Theorem 3.1 that the inherent Bellman error is a measure of the accuracy of function approximation under the Bellman operator corresponding to an

MDP. Thus, the varying values of $\mathcal{I}$ and $d$ in Figure 4 correspond to a natural setting where increasing the number of model parameters (i.e. increasing $d$) corresponds to an increase in the accuracy of function approximation (i.e. a decrease in $\mathcal{I}$). Thus the results reported in Figure 4 demonstrate that, even in the natural setting where increased model capacity leads to increased accuracy, there can be a complete reversal in the ordering of algorithm performance between the low and high-data regimes.

Figure 5 reports results on the number of samples required for training with the baseline algorithm that learns the state-action value distribution to reach the same performance levels achieved by the dueling algorithm for each individual MDP from ALE low-data regime benchmark. These results once more demonstrate that to reach the same performance levels with the dueling algorithm, the baseline algorithm that learns the state-action value distribution requires orders of magnitude more samples to train on. As discussed in Section 4.1, more complex representations for broader classes of distributions come at the cost of a higher sample complexity required for learning. One intriguing fact is that the original SimPLE paper provides a comparison in the low-data regime of their proposed algorithm with the Rainbow algorithm which is essentially an algorithm that is designed in the high-data regime by having the implicit assumption that the state-of-the art performance profile must transfer monotonically to the low-data regime. These instances of implicit assumptions also occur in DR$\mathcal{Q}^{\text{ICLR}}$, CURL, SPR and Efficient-Zero even when comparisons are made for more advanced algorithms such as MuZero.

**Datasets are Created and Founded on Implicit Assumptions:** Thus far we have discussed the pivotal role of implicit assumptions on the algorithmic comparisons and developing baselines in deep reinforcement learning. However, this issue further extends back to even how the entire Arcade Learning Environment 100K benchmark was established. The ALE 100K was initially created to allow researchers to work on a subset of games instead of full set of games used in the high-data regime (Kaiser et al., 2020), and this benchmark is currently used by any algorithm developed for the low data regime. However, the entire ALE 100K benchmark was in fact built on the selection bias of choosing games that performed better either with the proposed algorithm of the paper that proposed the entire benchmark, or with Rainbow which we extensively demonstrated throughout the paper is an algorithm that is subjected to the implicit assumption bias on monotonicity across regimes. Thus the issues we explicitly discover and analyze in our paper are not limited to baselines but further extend to entire benchmarks that we evaluate reinforcement learning algorithms on.

Our paper discovers that the canonical methodological choices made in a major line of deep reinforcement learning research that is based on these implicit assumptions, give incorrect signals on why and what makes these algorithms work when designed for the low-data regime, and hence affect future research directions while misdirecting research efforts from ideas that could have worked in the algorithm design process.

## 6 CONCLUSION

In this paper we aimed to answer the following questions: *(i) What are the canonical methodological choices that fundamentally effects the progress in deep reinforcement learning research*, *(ii) What is the underlying theoretical relationship between the performance profiles and sample complexity regimes?*, and *(iii) Do the performance profiles of deep reinforcement learning algorithms designed for certain data regimes translate monotonically to a different sample complexity region?* To be able to answer these questions we provide theoretical analysis on the sample complexity of the baseline deep reinforcement learning algorithms. We conduct extensive experiments both in the low-data regime 100K Arcade Learning Environment and high-data regime baseline 200 million frame training. Our analysis reveals that under the canonical methodological choices a major line of research resulted in suboptimal conclusions. In particular, both theoretical and empirical analysis provided in our paper demonstrate that the performance profiles of deep reinforcement learning algorithms do not have a monotonic relationship across sample complexity regimes. The underlying assumption of the monotonic relationship of the performance characteristics and the sample complexity regimes that is currently present in many recent state-of-the-art works led these studies to result in incorrect conclusions. Our paper demonstrates that several baseline $\mathcal{Q}$ algorithms perform better than a line of recent algorithms claimed to be the state-of-the-art.

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
