# OpenReview forum: "A Large-Scale Analysis on Methodological Choices in Deep Reinforcement Learning"
_ICLR.cc/2025/Conference — Submitted to ICLR 2025_

### Official Review · Reviewer_5jGi · 2024-10-21

**Soundness:** 1
**Presentation:** 2
**Contribution:** 3
**Rating:** 3
**Confidence:** 4

**Summary:**

The authors argue that the performance of RL algorithms in the high-data setting may not be indicative of the performance in the low-data setting. In particular, they emphasize the difference of distributional RL in the low and high data settings. They demonstrate this with experiments in the Atari 100k setting and theoretical results based on the linear function approximation setting.

**Strengths:**

I think that questioning our experimental settings is always a valuable contribution, and it’s nice to see more concrete evidence that the Atari 100k benchmark doesn’t necessarily generalize to the general Atari benchmark.

**Weaknesses:**

Clarity: The writing is extremely superfluous in many places and this makes it hard to follow the contributions and the arguments being made.

Overclaims:
- One claim in the paper is that the distributional RL may harm early learning, however, comparing C51 to Dueling is not a direct comparison, since it introduces a new technique. A fairer comparison would be C51 + Dueling to Dueling or C51 to DQN.
- I’m not sure I agree that the experiments are “large scale extensive” as the results omit several notable modern algorithms for Atari 100k (e.g., BBF, EfficientZero, DreamerV3) and only examine a single setting.
- Line 507 states “Our paper demonstrates that several baseline Q algorithms perform better than a line of recent algorithms claimed to be the state-of-the-art” but the results only show that Dueling DQN outperforms some other DQN-based algorithms designed for the full Atari setting, in the Atari 100k setting. But it doesn’t show Dueling outperforms actual SOTA algorithms for Atari 100k, nor does it show Dueling outperforms algorithms in the full Atari setting, so I’m not convinced by this claim.

**Questions:**

- It’s unclear to me what the incorrect conclusions made by prior work are exactly, could you expand on this point? (line 83)
- In Figure 1, shouldn’t dueling be attached to several of these methods, such as Rainbow, OTR, DER, and DrQ, which all use the dueling architecture?
- How much data is the high data regime results in Figure 1 (right) based on? Since these results don’t seem to correspond to the original paper results.
- Line 416 talks about the performance of DRQ vs Dueling, where are these numbers coming from? (Table or figure?)

---

> ### Author Response · Authors · 2024-11-23
> **Author Response**
>
> **1.** *”One claim in the paper is that the distributional RL may harm early learning, however, comparing C51 to Dueling is not a direct comparison, since it introduces a new technique. A fairer comparison would be C51 + Dueling to Dueling or C51 to DQN.”*
>
> ---
>
> Please note that this comparison is already currently present in our paper. DER is essentially C51 + Dueling, and the results in Figure 1 and Figure 5 demonstrate that indeed dueling substantially outperforms C51 + Dueling.
>
> ---
>
> **2.** *”How much data is the high data regime results in Figure 1 (right) based on? Since these results don’t seem to correspond to the original paper results.”*
>
> ---
>
> Figure 1 reports low data and high data regime results which correspond to 100K for low data and 200 million for high data. The results of Figure 1 (right) indeed correspond to the original paper results. This is also explained between line 33 and 38 in the introduction.
>
> ---
>
> **3.** *”Line 416 talks about the performance of DRQ vs Dueling, where are these numbers coming from? (Table or figure?)”*
>
> ---
>
> The results are reported in Figure 5 and this is explained starting from Line 375.
>
> ---
>
> **4.** *”In Figure 1, shouldn’t dueling be attached to several of these methods, such as Rainbow, OTR, DER, and DrQ, which all use the dueling architecture?”*
>
> ---
>
> Please read the caption of Figure 1 where it states *"A directed arrow from Algorithm A to Algorithm B means that the algorithm B provides comparison against the algorithm A as a baseline."*
>
> Thus, Figure 1 highlights that all of these methods failed to provide a comparison to the baseline dueling algorithm even though some of these methods were even built on top of it, read Line 412 and 415. Hence, a baseline algorithm that predates these recent algorithms in fact outperforms these recent algorithms designed for the low-data regime.
>
> ---
>
> **5.** *”Line 507 states “Our paper demonstrates that several baseline Q algorithms perform better than a line of recent algorithms claimed to be the state-of-the-art” but the results only show that Dueling DQN outperforms some other DQN-based algorithms designed for the full Atari setting, in the Atari 100k setting.“*
>
> ---
>
> This statement you make here is incorrect. The results reported in Figure 1 and Figure 5 further demonstrate that dueling outperforms a wide portfolio of algorithms designed for the low-data regime.
>
> ---
>
> **6.** *”It’s unclear to me what the incorrect conclusions made by prior work are exactly, could you expand on this point? (line 83)”*
>
> ---
>
> These incorrect conclusions are extensively explained in the empirical section of our paper. While there are many instances of these incorrect conclusions that are extensively discussed in the main body of our paper, here you can find an itemized form of one of these incorrect conclusions.
>
> I.  The DRQ paper uses the dueling algorithm and proposes a training method on top of the dueling algorithm.
>
> II.  The DRQ paper, although based on dueling, does not provide direct comparison to dueling.
>
> III. Our paper demonstrates that the baseline dueling algorithm performs better than the DRQ algorithm.
>
> Thus, an algorithm that is built on dueling and is proposed to improve performance, in fact lowers the performance of dueling. Thus, our paper explicitly investigates and analyzes these methodological choices that lead to incorrect conclusions.
>
> ---
>
> **7.** *”I’m not sure I agree that the experiments are “large scale extensive” as the results omit several notable modern algorithms for Atari 100k (e.g., BBF, EfficientZero, DreamerV3) and only examine a single setting.”*
>
> ---
>
> Please note that the algorithm proposed in BBF [1] is also SPR on top of Rainbow. Hence, this paper is also making the exact same implicit assumption that our paper describes, discusses and provides a deep concrete explicit analysis for.
>
> Furthermore, we currently mention in the main body of our paper that these implicit assumptions are also made in EfficientZero as well, and note that the implicit assumptions and the methodological choices still persistently appear in a large body of even very recent work [3].
>
> [1] Max Schwarzer, Johan Obando-Ceron, Rishabh Agarwal et al. Bigger, Better, Faster: Human-level Atari with human-level efficiency, ICML 2023.
>
> [2] Mastering Atari Games with Limited Data, NeurIPS 2021.
>
> [3] PLASTIC: Improving Input and Label Plasticity for Sample Efficient Reinforcement Learning, NeurIPS 2023.

---

> > ### Comment · Reviewer_5jGi · 2024-11-24
> >
> > Thank you for clarifying my questions, this is helpful for further understanding the paper. However, I don't feel as if any of my listed weaknesses has been addressed so I'm leaving my score as is.
> >
> > In response:
> > - 1. DER also uses priotization, noisy nets, n-step returns, and modifies a number of hyperparameters, so I would disagree with this statement.
> > - 7. It's unclear to me how building on top of Rainbow means BBF is methodological unsound. If the observation is that Distributional RL may be hurting the performance of modern algorithms in the low regime setting, then I suggest showing this result directly. If the observation is that dueling is a strong baseline, then I'm not sure this is pertinent for modern results, since BBF far outperforms dueling DQN.

---

> > > ### Author Response · Authors · 2024-11-26
> > > **Author Response**
> > >
> > > ---
> > >
> > > **1.** *”DER also uses priotization, noisy nets, n-step returns, and modifies a number of hyperparameters, so I would disagree with this statement.”*
> > >
> > > ---
> > >
> > > Please see Figure 5 of [1]. Figure 5 of [1] demonstrates that as n increases for n-step return performance significantly increases. Furthermore, please see [2]. The ablation studies reported in [2] demonstrates that indeed NoisyNetworks increases the performance. DER is built on top of C51 with these additional components, i.e. n-step return, NoisyNets and dueling, that in fact increases the performance of the C51 agent. Thus, the results reported in prior work clearly demonstrate that C51+dueling simply performs lower than DER.
> > >
> > > [1] Hado van Hasselt, Matteo Hessel, John Aslanides. When to use parametric models in reinforcement learning? NeurIPS 2019.
> > >
> > > [2] Rainbow: Combining Improvements in Deep Reinforcement Learning, AAAI 2018.
> > >
> > > ---
> > >
> > > **7.** *”It's unclear to me how building on top of Rainbow means BBF is methodological unsound. If the observation is that Distributional RL may be hurting the performance of modern algorithms in the low regime setting, then I suggest showing this result directly. If the observation is that dueling is a strong baseline, then I'm not sure this is pertinent for modern results, since BBF far outperforms dueling DQN.”*
> > >
> > > ---
> > >
> > > The results of our paper can be interpreted as it may be even possible to remove the distributional part in the BBF and obtain higher scores. However, the main point of our paper is to demonstrate the implicit assumptions on monotonicity across regimes and the resulting methodological choices made based on these assumptions that led to suboptimal conclusions in algorithmic choices.

---

> > > > ### Comment · Reviewer_5jGi · 2024-11-27
> > > >
> > > > > Furthermore, please see [2]. The ablation studies reported in [2] demonstrates that indeed NoisyNetworks increases the performance.
> > > >
> > > > The ablation studies reported in [2] are based on the 200M setting, so if your paper is correct, then we cannot rely on monotonicity across regimes to determine the performance of NoisyNetworks.
> > > >
> > > > > [...] the resulting methodological choices made based on these assumptions that led to suboptimal conclusions in algorithmic choices.
> > > >
> > > > If this is true, then it'd be best to show it. As is, it's only hypothetical that modern algorithms are harmed by these assumptions.

---

### Official Review · Reviewer_TBFr · 2024-10-24

**Soundness:** 3
**Presentation:** 1
**Contribution:** 3
**Rating:** 3
**Confidence:** 4

**Summary:**

Authors claim that conclusions of deep rl algorithms comparisons only hold for the data regime the algorithms were trained on. Hence, they compare distributional and classical deep q-learning on both the atari 100K and the atari 200M benchmarks. They infer that unlike what has been claimed by recent research performed in the low data regime (DRQ, Simple, ...) Dueling Q-Learning (Wang 2016) appear to be the regime-independent SOTA.

**Strengths:**

The main claim of the paper that conclusions about deep rl only holds for the data regime they are trained on is very smart and properly supported in the paper. Lots of experiments were conducted. This paper's claims are of great importance for the rl community.

**Weaknesses:**

I find this paper poorly written and poorly presented.

For me, the main weakness of the paper is that the main claim of the paper is so well supported by fig 1 (which is by the way repeated in fig 4) that adding all those proofs and learning curves in the main paper just kills its readibility and target audience which should be the whole RL community.

I think the proofs should go in the appendix and the experimental section needs to be rewritten.

In particular, baselines and environment could be described in distinct paragraphs which would prevent having so much acronyms. For example, why define DRQ^ICLR instead of just citing a paper ?

To rewrite the experimental section, you should have a look at the omitted empirical rl litterature. You can find some papers there : https://esraasaleh.com/cs-blog/experimental-rigour-in-machine-learning/ . I would particularly recommend the author to look at:
- Deep Reinforcement Learning at the Edge of the Statistical Precipice 2021
- Empirical Design in Reinforcement Learning 2023
- AdaStop: adaptive statistical testing for sound comparisons of Deep RL agents 2024

In particular, you can reproduce fig 5 of AdaStop: adaptive statistical testing for sound comparisons of Deep RL agents 2024 or fig 9 of Deep Reinforcement Learning at the Edge of the Statistical Precipice 2021 with some atari games in both the low and high data regimes.

**Questions:**

- Cite Arcade Learning Environment line 37.
- What does canonical mean in your introduction?
- Is it really true that people make the assumption of monotonic data regime (lines 40-46)? Maybe you should add citations.
- Missing citations for MDPs QRDQN, IQN, ... in the preliminary section.
- Can you confirm that your Human normalised scores are obtained by evaluating the learned policies (the Q-nets) outside of the training loops ?

---

> ### Author Response · Authors · 2024-11-23
> **Author Response**
>
> Thank you for writing that the main claim of the paper is very smart and properly supported and our paper's claims are of great importance for the RL community.
>
> ---
>
> **1.** *”For example, why define DRQ^ICLR instead of just citing a paper ?”*
>
> ---
>
> Because there is indeed a distinction between DRQ$^\text{ICLR}$ and  DRQ$^{\text{NeurIPS}}$, DRQ$^{\text{ICLR}}$ refers to the original paper [1] and DRQ$^{\text{NeurIPS}}$ refers to the reproduced version of the DRQ algorithm by [2]. This is also explained in our paper in footnote 1 in Page 8.
>
> [1] Image Augmentation Is All You Need: Regularizing Deep Reinforcement Learning from Pixels, ICLR 2021.
>
> [2] Deep reinforcement learning at the edge of the statistical precipice, NeurIPS 2021.
>
> ---
>
> **2.** *"Cite Arcade Learning Environment line 37.”*
>
> ---
>
> Currently, Arcade Learning Environment is already cited in Line 319 and 320. However, we can also cite in Line 37.
>
>
> ---
>
> **3.** *”Is it really true that people make the assumption of monotonic data regime (lines 40-46)? Maybe you should add citations.“*
>
> ---
>
> Yes, it is true. This is explained and cited in the main body of the paper in the empirical analysis section. However, we can also further cite here too.
>
> ---
>
> **4.** *”Can you confirm that your Human normalised scores are obtained by evaluating the learned policies (the Q-nets) outside of the training loops ?”*
>
> ---
>
> Yes.
>
> ---
>
> **5.** *”What does canonical mean in your introduction?”*
>
> ---
>
> We use canonical in the exact same meaning with the prior work [1].
>
> [1] Mastering Atari, Go, Chess and Shogi by Planning with a Learned Model, Nature 2020.

---

> > ### Comment · Reviewer_TBFr · 2024-11-24
> > **Not convinced.**
> >
> > Thank you your answer.
> > "Is it really true that people make the assumption of monotonic data regime (lines 40-46)? Maybe you should add citations.“
> > Well then can you please give an example?
> >
> > In the end I think this paper is too poorly written for anyone to understand without reading a lot of other references. Those references are also not cited where they should be making this paper too hard to understand outside of experts and has no place at ICLR in my opinion.

---

> > > ### Author Response · Authors · 2024-11-27
> > > **Author Response**
> > >
> > > Thank you for your response.
> > >
> > > As shown in Figure 1, a line of work starting from the creation of the ALE 100K benchmark, instances of which include, but are not limited to, SimPLe [1], OTR [2], DER [3], CURL [4], and DRQ [5], make these implicit assumptions on monotonicity. Thus, this line of work makes the implicit assumption that the algorithmic performance ranks across the regimes must be monotonic, i.e. Rainbow is the best performing algorithm in the high-data regime and thus, it must have the top ranked performance in the low-data regime, and benchmark against either the tuned version or Rainbow itself. However, our paper provides both theoretical and empirical analysis demonstrating that this implicit assumption is in fact not correct and algorithmic choices made on these implicit assumptions led research to incorrect and suboptimal conclusions.
> > >
> > > [1] Model-based reinforcement learning for atari. ICLR 2020.
> > >
> > > [2] When to use parametric models in reinforcement learning? NeurIPS 2019.
> > >
> > > [3] Do recent advancements in model-based deep reinforcement learning really improve data efficiency? 2020.
> > >
> > > [4] Curl: Contrastive unsupervised representations for reinforcement learning. ICML 2020
> > >
> > > [5] Image augmentation is all you need: Regularizing deep reinforcement learning from pixels. ICLR 2021.
> > >
> > > ---
> > >
> > > We are saddened to hear that you think our paper was too hard to understand outside the experts. We now made slight edits to Section 5 to improve readability. For making the paper more readable by everyone at ICLR, not only by the experts, we would also be happy to and willing to integrate any further concrete suggestion on the presentation of our paper.

---

> > > > ### Comment · Reviewer_TBFr · 2024-11-27
> > > > **Still a valid work**
> > > >
> > > > Don't get me wrong, I think your result about regime dependence in performance is interesting and more or less well-supported by empirical evidence. However you should rewrite the paper to make it way way clearer. propobably you should have more tables with like Algos | Dueling | ... | Data Regime | Target | Year to help people better situate existing work. ICML is right around the corner.

---

### Official Review · Reviewer_i3e2 · 2024-11-03

**Soundness:** 4
**Presentation:** 3
**Contribution:** 3
**Rating:** 6
**Confidence:** 5

**Summary:**

This paper provides a fundamental perspective, questioning the methodological and design choices of deep RL algorithms. It specifically focuses on value based deep RL algorithms and addresses questions around sample complexity and monotonic improvement of deep RL algorithms, in both the low data and high data regime. For experimental ablations, it uses the ALE benchmark, and presents new findings showing that an algorithm performance can vary in different regions of sample complexity; and claims that recent works may have been built from incorrect conclusions, since they implicitly assume that algorithm performance in asymptotic regime is a clear indicative that the same algorithm would also outperform in the low data regime. Finally, through careful empirical experiments, with supporting theoretical statements, this paper shows that some of the baseline Q algorithms from few years back, can still outperform model value based deep RL algorithms which claim to be state of the art.

**Strengths:**

In my opinion, this paper’s biggest strength is how carefully it chose to analyse some existing algorithms, and addresses really fundamental questions that provides new perspectives to commonly held beliefs. It is a well written paper with findings presented clearly; with supporting theoretical statements drawn from prior works. It is works like these that encourages other researchers in the community to often revisit fundamentals, even when the community is moving towards a different direction scaling up existing algorithms. Without such works, often common misconceptions will have a long lasting effect, incorrectly misleading recent works, while issues existing in fundamental algorithmic principles.

I would like to point out few key strengths of the paper :

1. The findings are presented carefully and the paper is well written (although can sound a bit repetitive in certain cases). Given the vast majority of value based or policy based algorithms, it is often hard to pick existing algorithms to do careful ablation study. As such, the authors pick value based RL algorithms, specifically Q algorithms from the starting of the deep RL era, and compares to recent algorithms that are claiming to be state of the art.

2. Past works have studied reproducibility or reliability issues in deep RL (Henderson et al., 2017) or even questioned about empirical reporting metrics often used for analysing performance of deep RL algorithms (Agarwal et al., 2021). While these works have raised issues around deep RL algorithms, often leading to the community being more aware of such issues - this work takes a different route compared to those and addresses a fundamental question : around sample complexity of deep RL algorithms, and what conclusions does the community draw looking at the asymptotic performance regime of modern RL algorithms.

3. It is easy to believe that when a deep RL algorithm performs well in high data or asymptotic regime, other works tend to build up from it quite easily, with the race towards state of the art. In value based deep RL algorithms, several works often compare to the 100k ALE benchmark, or the 200M training frames ALE benchmark - and only compares to existing baselines that seem promising in these benchmarks. The issue arises when these new algorithms are built from other fundamental Q algorithms, and since they already claim to outperform those prior baselines in the ALE benchmark - any new work would tend to ignore the prior baselines and only compare to the recent ones. This becomes exacerbated when performance is only compared to a particular sample complexity regime - often comparing asymptotic performance.
This work does a great job in pointing out that algorithm performance may vary depending on different sample complexity regimes. While past works may have implicitly been aware of this, e.g when analysing differences in performance curves - this was never formally studied through the lens of theoretical sample complexity and empirical performance. The findings in this work are clear and presents interesting insights.

4. Experimental results are well presented, and the figures provide clear insights on the different range of modern to older deep RL algorithms. One quick suggestion here for improvement would be to make the figure captions more self explanatory - ie, to include the conclusion/insight drawn from each result in the caption itself, without having to refer to the figure and look into the paper text for the conclusion drawn from that particular figure.

**Weaknesses:**

While the paper provides interesting insights and carefully presents findings and conclusions, there are few things maybe worth highlighting here.

1. Some of the text and findings in the paper often seem repetitive; maybe worth re-writing few parts of it.

2. I think the issue raised here are fundamental around deep Q algorithms, and not specifically to any distributional Q learning algorithms. Often the reference to distributional RL in the text seem confusing and presented as if the conclusions are only when comparing sample complexity regime for distributional Q algorithms. If I understand correctly, the paper presents the issues that are more generic to distributional RL - so it might be worth clearly stating that and not convoluting too much with distributional Q algorithms here. Please correct me if I’m wrong here.

3. Experimental comparisons are often made between Dueling, IQN, C51 and others - referring to the comment in (2), if we do not present results specifically to distributional Q algorithms, would it be possible to also present basic Q algorithms (like DQN for example, or some variant of DQN) in the figures here? This would help put the paper in more context to addressing how the common misconceptions around performance and sample complexity exists widely - and that very basic deep Q algorithms may in fact still be good depending on the sample complexity regime being analysed.

4. I think (also referring to Q3 below) - the bar for this paper can be increased significantly if some policy based algorithms are also introduced in the experiments for a comparison.

5. (Minor) : Figure 4 can be hard to understand and not clear what exact message it is trying to convey. It might be useful to make the caption more self explanatory.

**Questions:**

1. I do not intend to add a question here that raises the experimental requirement significantly; but if it is within the scope of the work - maybe one or two experimental results also introducing policy based algorithms in this context might make the paper really interesting. For example, if you pick basic PPO or TRPO for these ALE benchmarks - can we see how those algorithms also compare to these existing ones, in different sample complexity regimes?

2. Maybe outside the scope of the paper - but this issue around performance comparison in different sample complexity regime also exists between deep policy based and value based algorithms. Can the authors provide some results showing comparisons between value and policy based algorithms here?

---

> ### Author Response · Authors · 2024-11-23
> **Author Response**
>
> Thank you very much for dedicating your time and preparing a well-thought out review, and thank you for referring to our paper as:
>
> *It is works like these that encourage other researchers in the community to often revisit fundamentals and without such works, often common misconceptions will have a long lasting effect, incorrectly misleading recent works, while issues existing in fundamental algorithmic principles.*
>
> ---
>
> **1.** *”I think the issue raised here are fundamental around deep Q algorithms, and not specifically to any distributional Q learning algorithms. Often the reference to distributional RL in the text seem confusing and presented as if the conclusions are only when comparing sample complexity regime for distributional Q algorithms. If I understand correctly, the paper presents the issues that are more generic to distributional RL - so it might be worth clearly stating that and not convoluting too much with distributional Q algorithms here. Please correct me if I’m wrong here.”*
>
> ---
>
> This is a great suggestion. Yes, the presented issue is indeed more general, we can definitely state this more clearly.
>
> ---
>
> **2.** *”Experimental comparisons are often made between Dueling, IQN, C51 and others - referring to the comment in (2), if we do not present results specifically to distributional Q algorithms, would it be possible to also present basic Q algorithms (like DQN for example, or some variant of DQN) in the figures here? This would help put the paper in more context to addressing how the common misconceptions around performance and sample complexity exists widely - and that very basic deep Q algorithms may in fact still be good depending on the sample complexity regime being analysed.”*
>
> ---
>
> This is also another great suggestion. In Figure 1 we actually present a wider range of algorithms to demonstrate how the common misconceptions around performance and sample complexity exist widely.
>
> ---
>
> **3.** *”I do not intend to add a question here that raises the experimental requirement significantly; but if it is within the scope of the work - maybe one or two experimental results also introducing policy based algorithms in this context might make the paper really interesting. For example, if you pick basic PPO or TRPO for these ALE benchmarks - can we see how those algorithms also compare to these existing ones, in different sample complexity regimes?”*
>
> ---
>
> Upon your suggestion we in fact recognized and wanted to highlight that the issues that our paper raises on the impact of methodological choices, indeed seem to extend to policy based methods. In particular, let us look closer at the results: the PPO median in the high data regime is 298%, and dueling median is 172%; however, we see that in the low data regime the orders seem to be reversed and the median for PPO is 2% and median for dueling is 23%.

---

### Official Review · Reviewer_bB2o · 2024-11-04

**Soundness:** 2
**Presentation:** 3
**Contribution:** 2
**Rating:** 3
**Confidence:** 3

**Summary:**

The paper investigates the methodological approaches commonly adopted in deep reinforcement learning (DRL) and challenges several prevalent assumptions. The authors argue that many DRL studies implicitly assume that an algorithm's performance in high-data settings (where ample training data is available) will predict its performance in low-data settings (where data is limited). Authors claim that this assumption has led to methodological biases and suboptimal conclusions in the field. The study highlights that common methodological choices in DRL, particularly the assumption of a monotonic relationship between high-data and low-data performance, are flawed.

Through theoretical analysis and empirical testing, the paper demonstrates that performance profiles of DRL algorithms do not maintain a simple, predictable relationship between data regimes. Algorithms optimized for high-data performance do not necessarily perform well with limited data, and vice versa. The authors provide theoretical proofs and empirical results showing that simpler algorithms, such as the dueling network architecture, can outperform more recent methods in low-data scenarios.

Large-scale experiments on the Arcade Learning Environment (ALE) reinforce the theoretical findings, showing that simpler, established algorithms often outperform newer, more complex approaches under limited data conditions. The paper advocates for including these simpler baselines in performance comparisons to provide a more accurate picture of algorithm efficacy.

**Strengths:**

**Strengths:**

1) The paper presents a large-scale empirical analysis across multiple algorithms and data regimes, using the Arcade Learning Environment (ALE) benchmark.

2) In addition to empirical findings, the paper includes thorough theoretical analysis. The proofs related to non-monotonic performance across data regimes are well-founded and add a layer of depth to the study.

3) The authors identify and explain methodological biases resulting from high-data assumptions.

**Weaknesses:**

**Potential weaknesses:**

1) The empirical analysis is heavily focused on the Arcade Learning Environment (ALE), which, while popular, may not fully represent the broader range of tasks and environments encountered in deep reinforcement learning. A more diverse set of benchmarks, such as continuous control tasks or real-world scenarios, could make the conclusions more generalizable.
While the authors advocate for a paradigm shift in evaluating DRL algorithms, the paper does not discuss how its findings could be applied in practice. For instance, guidelines on designing experiments or adjusting evaluation metrics to avoid high-data biases would make the recommendations more actionable for researchers.

2) A recent paper has highlighted the phenomenon of data regime issues, offering valuable insights and prompting a reformulation of the research question in the paper [1].

3) I do not think it is accurate to say that algorithms developed for Atari100k (Data Efficient RL) are compared with those developed for Atari 200M (high-data regime) [2, 3, 4]. Please check the section "Case Study: The Atari 100k benchmark" in [5].

4) I believe the main takeaway of the paper is incorrect. It is clear that agents that perform well in the large data regime do not necessarily perform well compared to those developed for Atari100k. DER is essentially the same as the Rainbow agent, with different hyperparameters to perform well when data is scarce [5].

5) I would argue that choosing the right hyperparameters for the algorithms that learn the state-action value distribution (the ones introduced in the paper) will significantly affect their performance on Atari100k.

6) Why has the entire Atari100k suite been plotted? The Atari100k benchmark is composed of 26 games, but there are only 24 plots in Figure 1 (supplementary PDF). What is happening with the other games?

7) Figure 2 gives the impression that Dueling is always effective across all the games, but this is not true, as shown in Figure 1 (supplementary PDF). I suggest reporting the IQM metrics for all the games and including the learning curves as you are doing.

8) I do not see any positive benefits of Dueling in 10 games. Why is that?

9) It is difficult to draw conclusions from Table 3. The sample-efficient algorithms are missing. What if you tweak the hyperparameters for the other algorithms (C51/QRDQN/IQN)?

10) How do you define the learning rate and epsilon for the agent? I do not see this information in Table 1 (supplementary PDF).


**References:**
1. Obando-Ceron, Johan, et al. "On the consistency of hyper-parameter selection in value-based deep reinforcement learning." In Reinforcement Learning Conference.
2. Schwarzer, Max, Ankesh Anand, Rishab Goel, R. Devon Hjelm, Aaron Courville, and Philip Bachman. "Data-Efficient Reinforcement Learning with Self-Predictive Representations." In International Conference on Learning Representations.
3. Schwarzer, Max, Johan Samir Obando Ceron, Aaron Courville, Marc G. Bellemare, Rishabh Agarwal, and Pablo Samuel Castro. "Bigger, better, faster: Human-level atari with human-level efficiency." In International Conference on Machine Learning, pp. 30365-30380. PMLR, 2023.
4. Ye, Weirui, Shaohuai Liu, Thanard Kurutach, Pieter Abbeel, and Yang Gao. "Mastering atari games with limited data." Advances in neural information processing systems 34 (2021): 25476-25488.
5. Van Hasselt, Hado P., Matteo Hessel, and John Aslanides. "When to use parametric models in reinforcement learning?." Advances in Neural Information Processing Systems 32 (2019).
6. Agarwal, Rishabh, Max Schwarzer, Pablo Samuel Castro, Aaron C. Courville, and Marc Bellemare. "Deep reinforcement learning at the edge of the statistical precipice." Advances in neural information processing systems 34 (2021): 29304-29320.

**Questions:**

**Please read the comments and questions in the weaknesses section.**

---

> ### Author Response · Authors · 2024-11-23
> **Author Response**
>
> You have some critical misconceptions in your review. Some stem from missing the results reported in our paper, some stem from understanding prior work and some stem from missing critical details.
>
> ---
>
> **1.** *”I do not think it is accurate to say that algorithms developed for Atari100k (Data Efficient RL) are compared with those developed for Atari 200M (high-data regime) [2, 3, 4]. Please check the section "Case Study: The Atari 100k benchmark" in [5].”*
>
> ---
>
> You have a critical misconception here. The paper [5] is the DER algorithm, and Figure 1 in our paper reports results on the DER algorithm as well. The results reported in Figure 1 of our paper demonstrate that the baseline dueling algorithm outperforms the DER algorithm. Thus the implicit assumption is the fact that assuming that since Rainbow is the best performing algorithm in the high-data regime, specifically tuning Rainbow for the the low data regime, i.e. DER, must be the best performing algorithm in the low data regime as well. However, our results demonstrate that dueling outperforms both DER and DRQ.
>
> Furthermore, note that the papers you cite [2,3,4] indeed also compare against DER, and indeed carry the implicit assumption that our paper discovers, describes and provides a deep concrete explicit analysis for.
>
> [2] Max Schwarzer, Ankesh Anand, Rishab Goel et al. Data-Efficient Reinforcement Learning with Self-Predictive Representations, ICLR 2021.
>
> [3] Max Schwarzer, Johan Obando-Ceron, Rishabh Agarwal et al. Bigger, Better, Faster: Human-level Atari with human-level efficiency, ICML 2023.
>
> [4] Weirui Ye, Shaohuai Liu, Thanard Kurutach, Pieter Abbeel, Yang Gao. Mastering Atari Games with Limited Data, NeurIPS 2021.
>
> [5] When to use parametric models in reinforcement learning?, NeurIPS 2019.
>
>
> ---
>
>
> **2.** *”I believe the main takeaway of the paper is incorrect. It is clear that agents that perform well in the large data regime do not necessarily perform well compared to those developed for Atari100k. DER is essentially the same as the Rainbow agent, with different hyperparameters to perform well when data is scarce [5].”*
>
> ---
>
> The main takeaway of the paper is indeed correct. See Figure 1.
>
> Note that DER [5] is just an abbreviation for Data Efficient Rainbow, and it is in fact the Rainbow algorithm that has hyperparameters tuned for the low data regime. The paper [5] shows that in fact the Rainbow algorithm is the best performing algorithm in the low-data regime, the hyperparameters just need to be tuned correctly. Our results show that the simple dueling algorithm that has been proposed years before in fact outperforms DER in the low-data regime.
>
> ---
>
> **3.** *”I would argue that choosing the right hyperparameters for the algorithms that learn the state-action value distribution (the ones introduced in the paper) will significantly affect their performance on Atari100k.”*
>
> ---
>
> The hyperparameters are already tuned for the low-data regime.
>
> ---
>
> **4.** *”Figure 2 gives the impression that Dueling is always effective across all the games, but this is not true, as shown in Figure 1 (supplementary PDF).”*
>
> ---
>
> As already has been reported in Table 1, indeed the dueling algorithm outperforms across the entire Arcade Learning Environment 100K benchmark in median and 20th percentile.
>
> ---
>
> **5.** *”It is difficult to draw conclusions from Table 3. The sample-efficient algorithms are missing. What if you tweak the hyperparameters for the other algorithms (C51/QRDQN/IQN)?”*
>
> ---
>
> Hyperparameters of these algorithms are already tuned for the low data regime. The conclusion of Table 3 is clear. Table 1 reports results across the entire Arcade Learning Environment 100K benchmark. The results across the benchmark demonstrate that dueling performs better.
>
> ---
>
> **6.** *”While the authors advocate for a paradigm shift in evaluating DRL algorithms, the paper does not discuss how its findings could be applied in practice. For instance, guidelines on designing experiments or adjusting evaluation metrics to avoid high-data biases would make the recommendations more actionable for researchers.”*
>
> ---
>
> Please see the strengths part of the review of Reviewer i3e2 for how the findings of our paper are relevant to practice.
>
> ---
>
> **7.** *”A recent paper has highlighted the phenomenon of data regime issues, offering valuable insights and prompting a reformulation of the research question in the paper [1].”*
>
> ---
>
> This paper [1] is an empirical study on the hyper-parameter tuning of deep reinforcement learning. On the other hand, our paper is about methodological choices made in deep reinforcement learning based on implicit assumption of monotonicity of performance ranks across regimes.
>
> Our paper and the paper you cite are about completely different concepts.
>
> [1] Johan Obando-Ceron and João G.M. Araújo et al. On the consistency of hyper-parameter selection in value-based deep reinforcement learning, RLC 2024.

---

> ### Comment · Reviewer_bB2o · 2024-11-25
> **Author Response**
>
> Thank you for replying to some of my questions.
> I am not convinced by the narrative of the paper. For me, it is unclear and lacks sufficient evidence to support its claims. I suggest the authors reframe the paper to improve clarity.
>
> Questions:
>
> A. From the paper: *"One intriguing takeaway from the results provided in Table 1 and the Figure 5 is the fact that the simple base algorithm dueling performs 15% better than the DRQ (NeurIPS) implementation, and 11% less than the DRQ (ICLR) implementation."*
>
> 1. Why is the performance of DRQ (NeurIPS) lower than DRQ (ICLR) [1]? It should be the opposite [1], not?
> 2. Table 5 does not say anything about the comparison with DRQ.
>
> B. From the paper: *"The instances of the implicit assumption that the performance profile of an algorithm in the high-data regime will translate to the low-data regime monotonically appear in almost all of the studies conducted in the low-data regime."*
> >I disagree with this. I believe the RL community does not assume that agents performing well in the large-data regime are also optimal for the low-data regime. As I mentioned before, when evaluating performance on Atari100k, it is important not to make this assumption. Most of the baselines used for Atari100k differ from those used for 200M. If so, can you please provide more evidence?
>
> C. From the paper: *"...the results reported in Section 5 prove that the performance profile of an algorithm in the high-data regime does not monotonically transfer to the low-data regime."*
> >Sure, [2] shows this and links those findings to hyperparameter tuning. However, this does not mean that the community assumes RL agents trained on 200M are sufficiently sample-efficient to perform well with scarce data.
>
> D. From the paper: *"Our extensive large-scale empirical analysis demonstrates that a major line of research conducted in the past five years resulted in suboptimal conclusions. In particular, Figure 5 shows that a simple baseline dueling algorithm from 2016, by a canonical methodological choice was never included in the comparison benchmark due to the implicit assumption that appears in all of the recent line of research that we have discussed in detail in Section 3."*
> >I believe this is not an accurate conclusion. The Dueling result appears to be the same as in the DER [2] paper. The authors proposed some additional hyperparameters and made the Rainbow agent more sample-efficient. In the context of this paper, it is referred to as the Dueling agent. Indeed, agents like SPR and BBF all used Dueling. I think the authors should reframe the motivation of the paper, as some arguments lack sufficient support.
>
> *"Thus, we demonstrate that this baseline algorithm in fact performs much better than any recent algorithm that claimed to be better than the baselines, even including algorithms that are built on top of the dueling algorithm."*
> >This is not right at all. You are missing a lot of recent work on sample efficiency that has used Atari100k for this.
>
> E. *"Please see the strengths part of the review of Reviewer i3e2 for how the findings of our paper are relevant to practice..."*
> >I encourage the authors to add a section about guidelines on designing experiments or adjusting evaluation metrics to avoid high-data biases. This would be very valuable for the community.
>
> To sum up, for this paper to be presented at a top-tier AI conference, it requires more clarity regarding its motivations and contributions. I feel that the problem definition and logical approach in this paper are insufficient. Based on the current responses from the authors and the state of the paper, I cannot improve my rating.
>
> References:
> 1. Agarwal, Rishabh, Max Schwarzer, Pablo Samuel Castro, Aaron C. Courville, and Marc Bellemare. "Deep reinforcement learning at the edge of the statistical precipice." Advances in neural information processing systems 34 (2021): 29304-29320.
> 2. Johan Obando-Ceron and João G.M. Araújo et al. On the consistency of hyper-parameter selection in value-based deep reinforcement learning, RLC 2024.
> 3. Van Hasselt, Hado P., Matteo Hessel, and John Aslanides. "When to use parametric models in reinforcement learning?." Advances in Neural Information Processing Systems 32 (2019).

---

### Official Review · Reviewer_CmZM · 2024-11-04

**Soundness:** 3
**Presentation:** 2
**Contribution:** 3
**Rating:** 6
**Confidence:** 3

**Summary:**

One commonly studied benchmark is the Atari 100K benchmark in which algorithms are evaluated in low sample complexity regimes. Distributional variants of RL algorithms have shown high performance in the high data regime and have been transferred over to the low data regime of Atari 100K. This paper investigates the differences across different data regimes both from a theoretical and empirical perspective. There are two main theoretical results. The first shows that results are non-monotonic across regimes, and the second investigates the difficulties of learning distributions (e.g., in distributional RL). The empirical results show how distributional variants underperform non-distributional variants in the low-data (Atari 100K regime), which has implications for prior work in the literature that may have overlooked the impact of distributional RL.

**Strengths:**

I think this paper is decently written from a didactic point of view. The work itself does appear to be novel, both in terms of theory and experiments. The theoretical results on the data regimes and distribution learning seem interesting, insightful, and non-vacuous.  I think the community would be interested in these results, as they a) have implications for algorithm design and selection dependent on the data regime, and b) contain novel insights about several popular and well-cited algorithms for the Atari 100K benchmark. The experiments indeed serve to demonstrate the authors' point and are, by my assessment, sufficient.

I have mentioned several weaknesses. I might have given a higher score if these weaknesses were not present. I think this paper is acceptable because it has useful insights that are shown clearly, but there are flaws with the writing and paper itself that suggest that the work at present is not ready.

**Weaknesses:**

- The paper does not clearly state what the canonical methodological choices are. For a study titled "A LARGE-SCALE ANALYSIS ON METHODOLOGICAL
CHOICES IN DEEP REINFORCEMENT LEARNING", it is not clearly stated anywhere. Considering this is large-scale analysis of choices, I wonder what other choices are being investigated (e.g., outside of distributional learning). Figure 1 has "instances of methodological choices", but it's a bit unclear how algorithms are methodological choices. Clarity here would help.
- I did feel as though the question: "(i) What are the canonical methodological choices that fundamentally effects the progress in deep reinforcement learning research" was not adequately addressed or studied in the paper. In fact, they were not even listed.
- The paper lacks a related work section. This is fine per se, but there is no part of the paper that discusses the extent to which these insights are or not studied in prior work.

- Typo: "(i) What are the canonical methodological
choices that fundamentally effects the progress in deep reinforcement learning research"
	- "effects" -> "affects"

**Questions:**

- Given that the paper's title is "A LARGE-SCALE ANALYSIS ON METHODOLOGICAL
CHOICES IN DEEP REINFORCEMENT LEARNING", can you please list out all the methodological choices that were investigated?

---

> ### Author Response · Authors · 2024-11-23
> **Author Response**
>
> Thank you for stating that our paper is novel, both in terms of theory and experiments, and the theoretical results we provide on the data regimes are interesting, insightful, and non-vacuous, and furthermore our paper contains novel insights about several popular and well-cited algorithms for the Atari 100K benchmark with experiments  indeed serving to demonstrate the points presented in our paper.
>
> ---
>
> **1.** *“Given that the paper's title is "A LARGE-SCALE ANALYSIS ON METHODOLOGICAL CHOICES IN DEEP REINFORCEMENT LEARNING", can you please list out all the methodological choices that were investigated?”*
>
> ---
>
> Our paper investigates a series of methodological choices made under the assumption that an algorithm's performance must be monotonic across the data regimes, i.e. if an algorithm performs the best in the high-data regime it must be the best performing algorithm in the low-data regime. We see several instances of this in the case of establishing Rainbow as a baseline algorithm and developing an extensive line of algorithms for the low data regime. Although it is an intuitive methodological choice to assume that the performance across regimes will be monotonic, our work demonstrates that it is indeed non-monotonic, and a simpler baseline algorithm, e.g. dueling, that has been proposed and published years prior to Rainbow in fact performs much better than Rainbow.
>
> Furthermore, in an orthogonal axis of investigation our results demonstrate that some of the quite recent algorithms developed for the low-data regime that built on these existing prior baseline algorithms, i.e. dueling, did in fact not provide direct comparison to the algorithms that they were building on top of since the baseline algorithms were originally proposed for the high-data regime. Thus, when a direct comparison is made our results demonstrate that these recent low data regime algorithms in fact perform worse than the baseline algorithms that they were in fact built on.
>
> Furthermore, on an independent axis our paper also reveals another methodological choice is how the Arcade Learning Environment 100K benchmark itself was established, in which we directly quote: *“Specifically, for the final evaluation we selected games which achieved non-random results using our method or the Rainbow algorithm using 100K interactions.”* Hence, another critical methodological choice is that the entire low-data regime dataset was established based on the assumption that our paper discovers and provides an extensive analysis for.

---

> > ### Comment · Reviewer_CmZM · 2024-11-25
> > **Reviewer Response**
> >
> > Thanks you for your reply. I still am slightly confused on the list of methodological choices that were investigated. From your answer, I was not able to clearly understand what the methodological choices were. Even in the last paragraph of the response, I see one methodological choice as "how the Arcade Learning Environment 100K benchmark itself was established" followed by a quote that doesn't clarify what the methodological choice is. But I don't see a clear articulation of the what the methodological choice is.
> >
> > Could you please list this out more clearly? For example, even a bullet list of methodological choices (with an explanation of what the choice is and how it is methodological) would make this more clear.
> >
> > I maintain my score.

---

### Meta-Review · Area_Chair_Nyi6 · 2024-12-21

**Metareview:**

The paper attempts to shed some light on systematic issues in current RL research through some theoretical and experimental results. The main results indicate that the relative ordering of algorithm performances can reverse between high-data and low-data regimes, which is supported theoretically and empirically.

While the paper brings interesting insights, I find it to be not ready for publication yet. The writing especially was verbose and confusing at times, which often missed the mark of stating an important point in a straightforward manner. I highly suggest working on polishing the paper and tightening the claims, which can potentially turn this paper into a strong contribution for a future submission.

**Additional Comments On Reviewer Discussion:**

Reviewers pointed out the issues with writing as well as the limited diversity of task suites used (only focused on Atari environments) to justify the claims. Reviewers also found the empirical results lacking because the authors did not compare against modern algorithms designed for the Atari 100k benchmark.

---

### Decision · Program_Chairs · 2025-01-22

Reject